# Green Catalytic Conversion of Some Benzylic Alcohols to Acids by NiO₂ Nanoparticles (NPNPs) in Water

Abdel Ghany F. Shoair [1,2,*], Mai M. A. H. Shanab [3], Nasser A. El-Ghamaz [4], Mortaga M. Abou-Krisha [5,6], Sayed H. Kenawy [6,7] and Tarek A. Yousef [6,8]

[1]   Department of Science and Technology, University College-Ranyah, Taif University, Taif 21975, Saudi Arabia
[2]   High Altitude Research Center, Taif University, Taif 21975, Saudi Arabia
[3]   Department of Chemistry, College of Sciences and Humanities Studies (Girls Section), Hawtat Bani Tamim, Prince Sattam Bin Abdulaziz University, Al-Kharj 11149, Saudi Arabia; m.hassan@psau.edu.sa
[4]   Department of Physics, Faculty of Science, Damietta University, New Damietta 34517, Egypt; elghamaz@du.edu.eg
[5]   Faculty of Science, Chemistry Department, South Valley University, Qena 83523, Egypt; mmaboukrisha@imamu.edu.sa
[6]   Chemistry Department, College of Science, Imam Mohammad Ibn Saud Islamic University (IMSIU), Riyadh 11623, Saudi Arabia; skibrahim@imamu.edu.sa (S.H.K.); tayousef@imamu.edu.sa (T.A.Y.)
[7]   Refractories, Ceramics and Building Materials Department, National Research Centre, El-Buhouth St., Dokki, Giza 12622, Egypt
[8]   Toxic and Nacrotic Drugs Laboratory, Department of Forensic Medicine, Mansoura Laboratory, Medico Legal Organization, Ministry of Justice, Mansoura 35516, Egypt
*   Correspondence: afshaair@tu.edu.sa

**Abstract:** The aqueous basic systems NiSO₄·6H₂O/K₂S₂O₈ (pH = 14) and NiSO₄·6H₂O/KBrO₃ (pH = 11.5) were investigated for the catalytic conversion of benzyl alcohol and some para-substituted benzyl alcohols to their corresponding acids in 75–97% yields at room temperature. The active species was isolated and characterized using scanning electron microscopy (SEM), transmission electron microscopy (TEM), X-ray powder diffraction, EDX and FT-IR techniques, and identified as comprising NiO₂ nanoparticles (NPNPs). The SEM and TEM images of the Ni peroxide samples showed a fine spherical-like aggregation of NiO₂ molecules with a nearly homogeneous partial size and confirmed the aggregation's size to be in the range of 2–3 nm. The yields, turn over (TO) and turn over frequencies (TOF) were calculated. It was noticed that the aromatic alcohols containing para-substituted electron donation groups gave better yields than those having electron withdrawing groups. The optimum conditions for this catalytic reaction were studied using benzyl alcohol as a model. The mechanism of the catalytic conversion reaction was suggested, in which the produced NPNPs convert alcohols to acids in two steps through the formation of the corresponding aldehyde. The produced NiO because of this conversion is converted again to NPNPs by the excess of K₂S₂O₈ or KBrO₃. This catalytic cycle continues until all of the substrate is oxidized.

**Keywords:** nickel peroxide nanoparticles (NPBPs); catalyst; benzyl alcohol; green oxidation

## 1. Introduction

The green protocol for the production of valuable chemicals is of great significance in the sustainable development of green economy [1]. The oxidation of benzyl alcohols to aromatic acids using green routes is very important to avoid bad effects on the environment [2]. The production of aromatic acids is important in organic synthesis, since it is used as an intermediate in the synthesis of many fine chemicals such as drugs and vitamins [3]. The use of stoichiometric amounts of inorganic transition metal salts such as copper (I) [4], copper(II) [5] permanganate [6], chromium(VI) [7] and iron(III) [8,9] is often toxic and generates a considerable amount of by-product. The use of precious metals such as Pt [10], Pd [11] and Ru [12] is industrially restricted due to their high cost and

scarcity. The extremely small size of nanoparticles enlarges the surface area and allows more reactions to occur, displaying high catalytic activity [13–17]. Recently, benzoic acid was synthesized via the pyrolysis of xylan to toluene followed by the subsequent catalytic oxidation of toluene to benzoic acid by the bimetallic catalyst $Co_2MnO_4$@MCM-41 [18]. Oroujzadeh et al. [19] reported the catalytic oxidation of benzyl alcohol to benzoic acid by the $[CuClL_2]$ (L = N-nicotinyl;N-N″-bis(tert-butyl) phosphoric triamide) complex at 78 °C in aceto-nitrile as a solvent. A mixture of benzaldehyde and benzoic acid was obtained from the catalytic oxidation of benzyl alcohol by the potassium salt of the Keggin heteropolyacids $K_5PW_{11}NiO_{39}$ in the presence of $H_2O_2$ [20]. To the best of my knowledge, few reports have been on for the applications of nickel peroxide in the catalytic oxidation of organic compounds. Goerge et al. reported a good review on the use of nickel peroxide for the oxidation of many organic compounds, including alcohols, phenols and amines [21]. The synthesis of diacetone-2-ketoL-gulonic acid, an intermediate in the synthesis of vitamin C, was reported by Weijlard via the addition of nickel salts in a solution of sodium hypochlorite [22]. Nakgawa et al. reported on the use of an aqueous solution of nickel-sulfate-treated sodium hypochlorite in an alkaline medium for the stoichiometric oxidation of primary alcohols to their corresponding acids and to their corresponding carbonyl compounds in an organic solvent [23], and also on the synthesis of 2-phenylbenzoxazoles from their corresponding Schiff bases [24,25]. As part of our ongoing interest in the catalytic oxidation of organic compounds by transition metal–oxo complexes [26], we decided to study the catalytic oxidation of a number of aromatic alcohols to their corresponding acids catalyzed by the reagent $NiSO_4 \cdot 6H_2O/K_2S_2O_8$ or $KBrO_3$ in an aqueous basic medium. A number of factors were studied to optimize the reaction conditions. The active species was isolated and characterized as comprising nickel peroxide nanoparticles (NPNPs). A plausible mechanism for this catalytic reaction was suggested.

## 2. Results and Discussion

### 2.1. Synthesis of the Catalyst

The nickel peroxide nanoparticles were synthesized using the co-precipitation method by mixing two equal amounts of nickel sulfate and potassium persulfate in the presence of four equivalent parts of KOH in water according to Equation (1):

$$NiSO_4 \cdot 6H_2O + 4KOH + K_2S_2O_8 \rightarrow NiO_2 + 3K_2SO_4 + 8H_2O \tag{1}$$

$K_2S_2O_8$ and $KBrO_3$ act as co-oxidants and react with $NiSO_4 \cdot 6H_2O$ in the presence of KOH as a basic medium to generate $NiO_2$, for example in Equation (1). The black nickel peroxide NPNPs were filtered, washed with $H_2O$ and dried at 60 °C for three hours.

### 2.2. Characterization of Nickel Peroxide Nanoparticles (NPNPs)

#### 2.2.1. FTIR Spectrum of $NiO_2$ Nanoparticles (NPNPs)

The FT-IR spectrum showed several significant absorption peaks (Figure 1). The absorption bands in the range of 500–550 $cm^{-1}$ were assigned to the NiO stretching vibrational mode [27,28]. The weak absorption band at around 720 $cm^{-1}$ is characteristic of the peroxo O-O stretching vibrational mode [29]. It was noticed that a broad absorption band and a weak band at 3405 $cm^{-1}$ and 1630 $cm^{-1}$ were attributable to O–H and H-O-H bending and stretching vibrational modes, respectively [30].

These bands were due to the hydration of the FTIR sample disk when it was prepared in open air. Other bands were observed in the range of 1070–1450 $cm^{-1}$; these bands were assigned to the absorbed carbon dioxide by the sample disk [31].

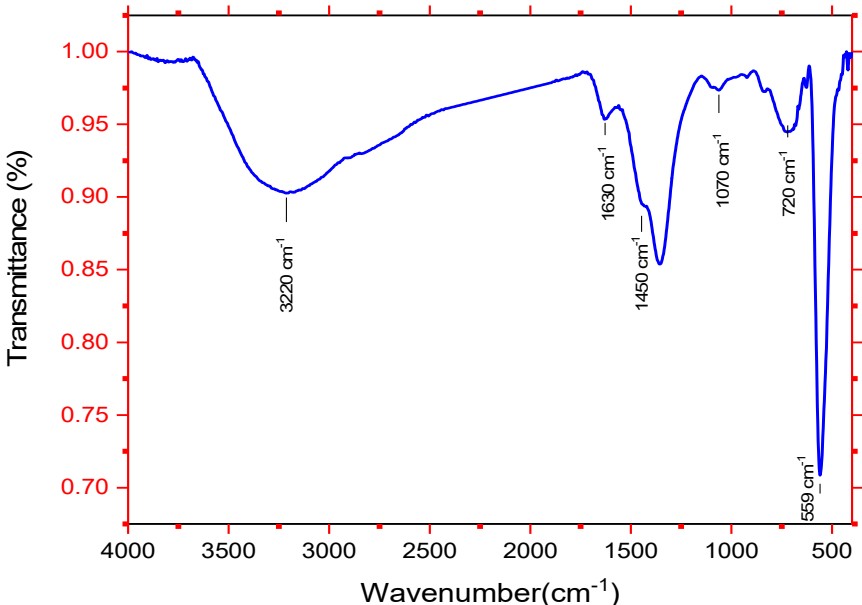

**Figure 1.** Vibrational Fourier transform infrared (FT-IR) spectrum of the $NiO_2$ nanoparticles (NPNPs).

### 2.2.2. EDX Analysis

The stoichiometry of the as-synthesized Ni peroxide ($NiO_2$) powder was examined using energy-dispersive spectroscopy (EDS), as shown in Figure 2. The atomic weight ration of nickel to oxygen (Ni/O) was found to be 1.96, which was in good agreement with the theoretically calculated value of 1.83 [32]. The results reported by Kooti et al. [28] showed that the experimental obtained values of Ni/O were 1.73 and 3.67 for $NiO_2$ and NiO, respectively. The present results extracted from the EDS confirmed the good stoichiometry of the prepared Ni peroxide.

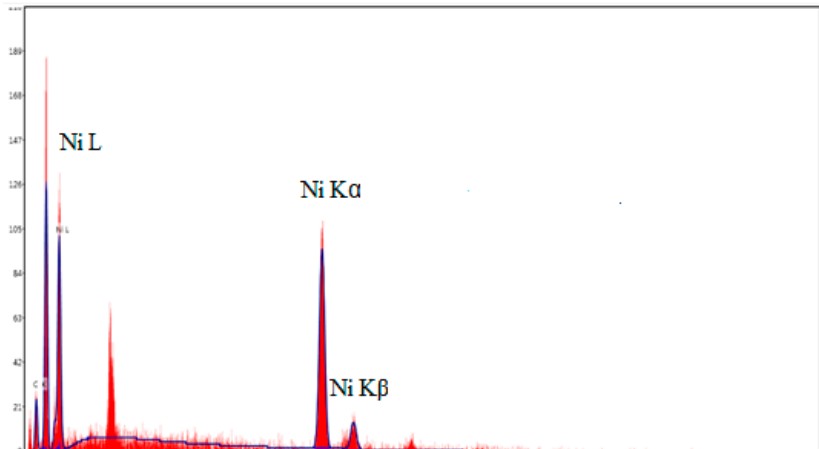

**Figure 2.** EDX spectrum of $NiO_2$ nanoparticles (NPNPs).

### 2.2.3. XRD Analysis

The XRD pattern of the $NiO_2$ powder is shown in Figure 3. The two broad peaks around 2θ ~ 17.6° and 37.2° with the superimposed sharp peaks indicate polycrystalline ultra-nano-sized particles of $NiO_2$. The peaks positioned at 2θ ≈ 37.8° and 43.9 can be assigned to the (111) and (200) crystal planes, respectively, (JCPDS 47-1049) [33]. The average crystallite size of the $NiO_2$ powder under investigation was found to be 1.3 nm. The obtained average crystallite size suggests the synthesized $NiO_2$ to comprise ultra-small nanoparticles. Earlier, Seguin et al. [32] reported and confirmed the monoclinic crystal

system of $NiO_2$. The average crystallite size (L) of $NiO_2$ powder can be estimated by employing the following Sherres's equation [33–35]:

$$L = \frac{k\lambda}{\beta \cos(\theta)}$$

where $k = 0.95$ is the wavelength of the used X-ray, $\theta$ is the full width at half maximum of the diffraction peak measured in radians, and $\theta$ is the Bragg's angle.

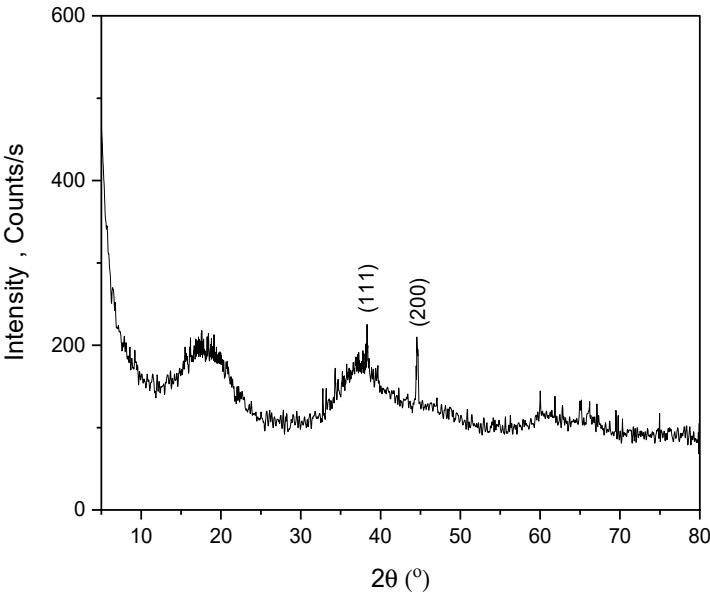

**Figure 3.** Powder X-ray diffraction (XRD) results for the $NiO_2$ nanoparticles (NPNPs).

### 2.3. SEM and TEM

The SEM images of $NiO_2$ samples are presented in Figure 4a,b. The SEM image in Figure 4a shows fine spherical-like aggregations of $NiO_2$ molecules with a nearly homogeneous particle size. The TEM image in Figure 4b confirms the aggregations' size to be in the range of 2–3 nm. This result confirms the ultra-small size of the $NiO_2$ nanoparticles obtained using Sherrer's equation. On the other hand, the electron diffraction pattern (EDP) in the inset of Figure 4b confirms the fine-sized polycrystalline phase of the $NiO_2$.

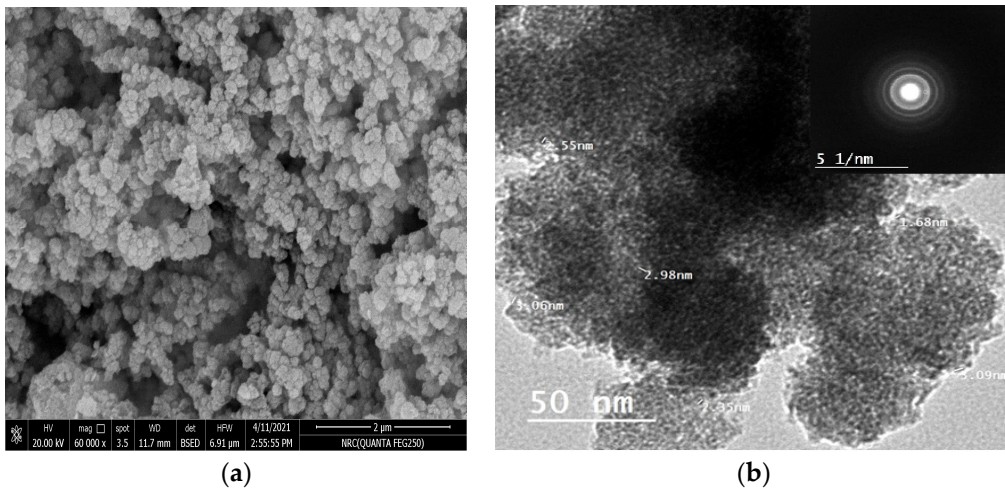

(**a**)                                          (**b**)

**Figure 4.** (**a**) SEM image of the NPNPs and (**b**) TEM images of the $NiO_2$ nanoparticles (NPNPs).

### 2.4. Catalytic Oxidation of Alcohols

We decided to investigate the catalytic activity of nickel peroxide nanoparticles (NPNPs) generated in situ from nickel salts such as $NiSO_4 \cdot 6H_2O$, $NiCl_2 \cdot 6H_2O$, and $Ni(CH_3COO)_2 \cdot 2H_2O$ in the presence of $K_2S_2O_8$ in 1.0 M aqueous KOH or $KBrO_3$ in 1.0 M aqueous $K_2CO_3$ for the catalytic conversion of benzyl alcohol and some *p*-substituted benzyl alcohols containing electron-withdrawing and electron-donating groups to their corresponding acids. Additionally, we also investigated the optimum conditions for conducting this catalytic reaction (Scheme 1).

$$R= H, CH_3, OCH_3, OH, NH_2, CHO, CN, NO_2 \text{ and } F_3C$$

**Scheme 1.** Catalytic oxidation of benzyl alcohol and some *p*-substituted benzyl alcohol by $NiSO_4 \cdot 6H_2O/K_2S_2O_8$.

Benzyl alcohol was selected as a model substrate of $NiSO_4 \cdot 6H_2O$ for the in situ generation of nickel peroxide nanoparticles (NPNPs), which were the active catalytic species in this reaction, with two co-oxidants $K_2S_2O_8$ in 1.0 M KOH (pH = 14) and $KBrO_3$ in 1.0 M $K_2CO_3$ (pH = 11.5) as co-oxidants.

A set of experiments were carried out using two different nickel salts instead of $NiSO_4 \cdot 6H_2O$ with two different co-oxidants additionally, and two experiments were performed at 50 °C and 80 °C. The results are presented in Table 1.

It was found that the oxidation of benzyl alcohol (10 mM) to the corresponding benzoic acid took place smoothly in a short time (4 h) with the use of 0.2 mM of $NiSO_4 \cdot 6H_2O$ (50 fold-excess of the substrate) and 20 mM of the co-oxidants (two fold-excess of the substrate) in 100 mL of aqueous KOH (1.0 M). These conditions gave 97% and 75% benzoic acid with $K_2S_2O_8$ and $KBrO_3$ as the co-oxidants, respectively, at ambient temperature (entry 1, Table 1). The yields obtained with $KBrO_3$ in 1.0 M $K_2CO_3$ (pH = 11.5) were lower than those obtained with $K_2S_2O_8$ in 1.0 M KOH (pH = 14), which was probably due to $K_2S_2O_8$ having a stronger oxidizing power and higher aqueous solubility and stability suitable for the formation of the active species (NPNPs) than $KBrO_3$.

When we conducted two reactions at 50 °C and 80 °C, respectively, the yields of the benzoic acid were lower than those obtained at ambient temperature and the reaction was smelly with the formation of unidentified products. These products result from the reaction of unreacted alcohol with the acid produced when the reaction is performed at these higher temperatures, and they were difficult to isolate and identify (entries 2 and 3, Table 1).

Under the previously established reaction conditions, the oxidation of benzyl alcohol was also carried out using different nickel salts such as $NiCl_2$ and $Ni(CH_3COO)_2 \cdot 2H_2O$ instead of $NiSO_4 \cdot 6H_2O$ (entries 4 and 5, Table 1). It was noticed that the oxidation was quite slow and a considerably lower amount of benzoic acid was obtained. The differences in the yields obtained when the reactions were performed with the different nickel salts reflected their differences in solubility, i.e., a higher product yield was obtained with $NiSO_4 \cdot 6H_2O$ than with $NiCl_2 \cdot 2H_2O$ and $Ni(CH_3COO)_2 \cdot 2H_2O$ because $NiSO_4 \cdot 6H_2O$ is more soluble in the reaction medium than the other salts.

On the other hand, a blank experiment was conducted in the absence of $NiSO_4 \cdot 6H_2O$, whereby the reaction did not proceed to completion and the acid was not detected (entry 6, Table 1). We separated the produced grey-green nickel oxide powder at the end of the reaction via filtration and washing with deionized water, recycled three times (entries 7, 8, and 9, Table 1) to produce nickel peroxide NPNPs, again with either excess $K_2S_2O_8$ in 1.0 M KOH or $KBrO_3$ in 1.0 M $K_2CO_3$, and used this for further catalytic oxidation reactions of benzyl alcohol to the corresponding acid benzoic, giving 80%, 76%, and 60% yields, respectively.

**Table 1.** Optimization of the reaction conditions.

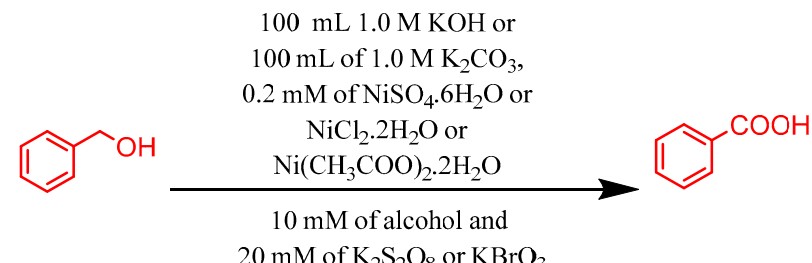

100 mL 1.0 M KOH or
100 mL of 1.0 M $K_2CO_3$,
0.2 mM of $NiSO_4.6H_2O$ or
$NiCl_2.2H_2O$ or
$Ni(CH_3COO)_2.2H_2O$

10 mM of alcohol and
20 mM of $K_2S_2O_8$ or $KBrO_3$

| Entry | Co-Oxidant | Y (%) | TO | TOF ($h^{-1}$) |
|---|---|---|---|---|
| 1 | $K_2S_2O_8$ | 97 | 48.5 | 12.13 |
|   | $KBrO_3$ | 75 | 36 | 9 |
| 2 | $K_2S_2O_8$ | 85 | 42.5 | 10.63 |
|   | $KBrO_3$ | 60 | 30 | 7.5 |
| 3 | $K_2S_2O_8$ | 81 | 40.5 | 10.13 |
|   | $KBrO_3$ | 63 | 31.5 | 7.9 |
| 4 | $K_2S_2O_8$ | 70 | 35 | 8.75 |
|   | $KBrO_3$ | 54 | 27 | 6.75 |
| 5 | $K_2S_2O_8$ | 50 | 25 | 6.25 |
|   | $KBrO_3$ | 50 | 25 | 6.25 |
| 6 | $K_2S_2O_8$ | 0 | 0 | 0 |
|   | $KBrO_3$ | 0 | 0 | 0 |
| 7 | $K_2S_2O_8$ | 80 | 40 | 10 |
|   | $KBrO_3$ | 70 | 35 | 8.75 |
| 8 | $K_2S_2O_8$ | 76 | 38 | 9.5 |
|   | $KBrO_3$ | 55 | 27.5 | 6.88 |
| 9 | $K_2S_2O_8$ | 60 | 30 | 7.5 |
|   | $KBrO_3$ | 40 | 20 | 5 |

Reaction conditions: All reactions were carried out at ambient temperature except for entries 2 and 3, which were carried out at 50 °C and 80 °C, respectively, and entries 4 and 5, where 0.2 mM of $NiCl_2·H_2O$ or 0.2 mM of $Ni(CH_3COO)_2·2H_2O$ was used instead of $NiSO_4·6H_2O$, respectively. Reaction time = 4 h; yield (Y) (%) = number of moles of produced acid × 100/number of moles of alcohol; turn over (TO) = number of moles of product/number of moles of catalyst; turn over frequency (TOF) = number of moles of product/number of moles of catalyst per hour.

We noticed the yields obtained in the recycling of the catalyst were less than those obtained the first time, probably due to the loss of some of the catalyst during the reaction and also the loss of its active sites.

The effect of the *p*-derivatives on the yield of the acid was studied via the oxidation of some *p*-substituted electron-donating groups (*p*-CH$_3$, *p*-CH$_3$O, OH, and NH$_2$) and *p*-substituted electron-withdrawing groups (*p*-CHO, *p*-NO$_2$, *p*-CN, and *p*-CF$_3$). We found that the yields obtained with the electron-donating substituents (entries 10, 11, 12, and 13, Table 2) were higher than those obtained with the electron-withdrawing groups (entries 14, 15, 16, and 17, Table 2). The possible reason for these observations was the fact that the electron-donating groups activate the ring, thereby enhancing the oxidation of alcohol into the corresponding acid, while the electron-withdrawing groups deactivate the phenyl ring and in this way retard the reaction. The convincing evidence was the fact that the yield obtained with the first group was higher than that obtained with the latter.

**Table 2.** The scope of the catalytic oxidation of benzyl alcohols to benzoic acids by $NiSO_4 \cdot 6H_2O/K_2S_2O_8$.

KOH 100 mL 1.0 M
$K_2CO_3$ 100 mL 1.0M
$NiSO_4 \cdot 6H_2O$ or $NiCl_2 \cdot 2H_2O$ or $Ni(CH_3CO_2)_2 \cdot 2H_2O$ (0.2 mM)
Alcohol (10 mM), $K_2S_2O_8$ or $KBrO_3$ (20 mM)

(R = CH₃, OCH₃, OH, NH₂, CHO, CN, NO₂ and F₃C) and piperonyl alcohol

| Entry | Substrate | Product | Co-Oxidant | Y (%) | TO | TOF ($h^{-1}$) |
|---|---|---|---|---|---|---|
| 10 | | | $K_2S_2O_8$ | 98 | 49 | 19.25 |
| | | | $KBrO_3$ | 70 | 35.5 | 8.6 |
| 11 | | | $K_2S_2O_8$ | 97 | 48.5 | 12.13 |
| | | | $KBrO_3$ | 63 | 31.5 | 7.5 |
| 12 | | | $K_2S_2O_8$ | 95 | 47.5 | 11.88 |
| | | | $KBrO_3$ | 65 | 32.5 | 7.9 |
| 13 | | | $K_2S_2O_8$ | 95 | 47.5 | 11.88 |
| | | | $KBrO_3$ | 66 | 33 | 8.25 |
| 14 | | | $K_2S_2O_8$ | 70 | 35.5 | 8.88 |
| | | | $KBrO_3$ | 40 | 20 | 5 |
| 15 | | | $K_2S_2O_8$ | 60 | 30 | 7.5 |
| | | | $KBrO_3$ | 30 | 15 | 3.75 |
| 16 | | | $K_2S_2O_8$ | 60 | 30 | 7.5 |
| | | | $KBrO_3$ | 30 | 15 | 3.75 |
| 17 | | | $K_2S_2O_8$ | 55 | 27.5 | 6.88 |
| | | | $KBrO_3$ | 25 | 12.5 | 3.13 |
| 18 | | | $K_2S_2O_8$ | 85 | 42.5 | 10.6 |
| | | | $KBrO_3$ | 50 | 25 | 6.25 |

However, we noticed that increasing the reaction time more than four hours (Figure 5a) and the amount of co-oxidants ($K_2S_2O_8$ and $KBrO_3$) more than 20 mM (Figure 5b) did not improve the yield.

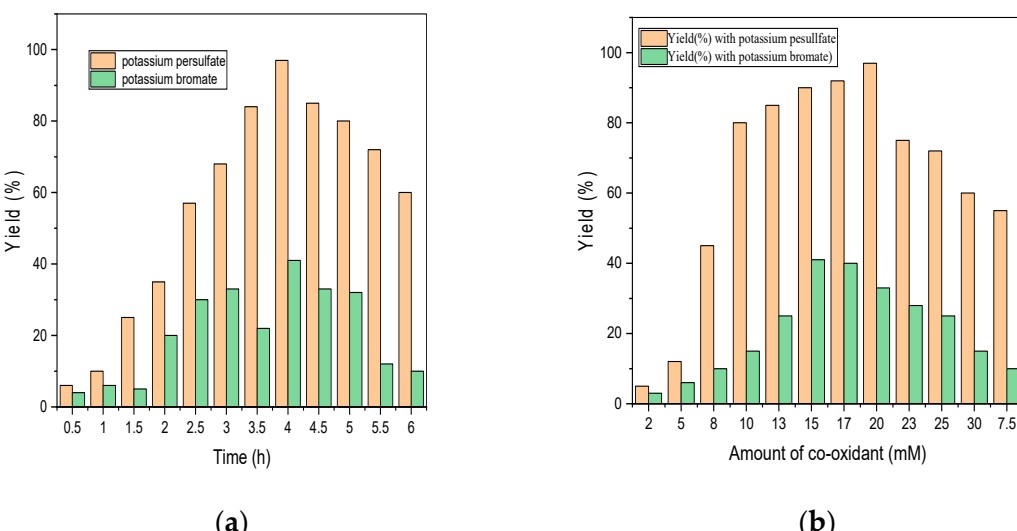

(a) (b)

**Figure 5.** (**a**) The time (h) against the yield (%) and (**b**) the amount of co-oxidant against the yield (%).

The reaction was self-indicating, whereby the green color of the reaction mixture changed to black upon the addition of potassium persulfate due to the formation of nickel peroxide nanoparticles (NPNPs). This color disappeared gradually with the formation of the green nickel oxide and completely disappeared when the potassium persulfate was consumed in the oxidation of the substrate.

However, this catalytic oxidation reaction is considered to be green because the reaction is selective and catalytic (alcohol is converted mainly to acid), the solvent used (water) is eco-friendly, and the oxidation is carried out at ambient temperature. However, our results are comparable with some recently reported protocols [36,37] for the catalytic oxidation of benzyl alcohols. It appears to be superior to most previously reported approaches. In our protocol, the yields of the acids were higher, and the reactions were conducted at room temperature with shorter reaction times than those reported by Han [38]. Recently, benzyl alcohol was catalytically oxidized to benzoic acid by the photocatalyst NH$_2$-MIL-125 (Ti) MOF in ethyacetate as a solvent and with O$_2$ as co-oxidant. In our recipe, we used water as a solvent, which is environmentally better than ethyl acetate, for which the reaction is longer than ours [39]. The catalyst, Ni@C/TiO$_2$-Z, was hydrothermally prepared and used for the oxidation of benzyl alcohol to benzoic acid at 96% in ethylacetate as a solvent for 18 h (a longer reaction time than our protocol) [40].

Finally, this protocol did not use any toxic solvent or chemical and was selective (the only product is the acid, with water as the sole by-product).

### 2.5. Mechanism of the Catalysis

It is very meaningful to trace the reaction process of the oxidation of alcohol to benzoic acid (BzCOOH) to understand the mechanism based on time-dependent curves (the percentages of BzOH, BzCHO, and BzCOOH with the reaction times) (Figure 6).

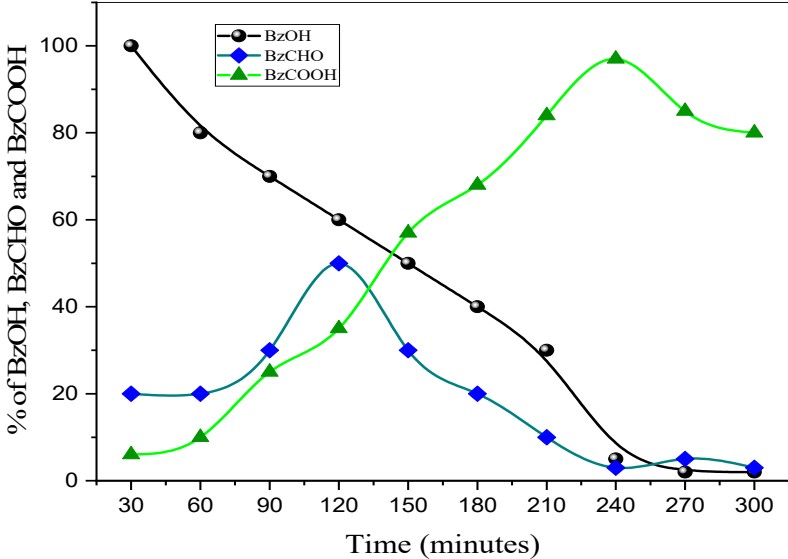

**Figure 6.** Time-dependent curves, showing the percentages of BzOH, BzCHO, and BzCOOH and the times in minutes.

Firstly, benzaldehyde was produced as a mid-product with consumption of half the amount of NiO$_2$ (first step) (Equation (2)):

$$NiO_2 + PhCH_2OH \rightarrow NiO + PhCHO + H_2O \tag{2}$$

Next, in the second step, the produced amount of benzaldehyde (BzCHO) decreased gradually due to its conversion to benzoic acid (BzCOOH) (Equation (3)):

$$NiO_2 + PhCHO \rightarrow NiO + PhCOOH \tag{3}$$

In this second step, the other half of the $NiO_2$ was completely consumed; thus, stoichiometrically, two moles of $NiO_2$ reacted with one of benzyl alcohol (BzOH) to produce one mole of benzoic acid (BzCOOH) according to the following Equation (4).

$$2NiO_2 + PhCH_2OH \rightarrow 2NiO + PhCOOH + H_2O \tag{4}$$

For this reason, the amount of the co-oxidant ($K_2S_2O_8$) was in two-fold excess of the benzyl alcohol (BzOH) and fifty-fold excess of the nickel salt. This means two oxidation steps occur at the same rate; therefore, we expect that the oxidation of benzyl alcohol (BzOH) to benzaldehyde (BzCHO) represents the rate-determining step [39].

However, the mechanism of this reaction occurred through the hydrogen abstraction mechanism, where benzyl alcohol (BzOH) coordinated to the nickel peroxide to form an unstable intermediate complex, which underwent intramolecular arrangement to produce the corresponding benzaldehyde (BzCHO) and nickel oxide. The produced nickel oxide reacted again with the excess of $K_2S_2O_8$ to produce nickel peroxide (Equation (5)):

$$NiO + K_2S_2O_8 + 2KOH \rightarrow NiO_2 + 2K_2SO_4 + H_2O \tag{5}$$

This product similarly reacted with the produced benzaldehyde (BzCHO) to produce benzoic acid (BzCOOH). This catalytic cycle continued until the benzyl alcohol (BzOH) and benzaldehyde BzCHO) were completely consumed (Scheme 2).

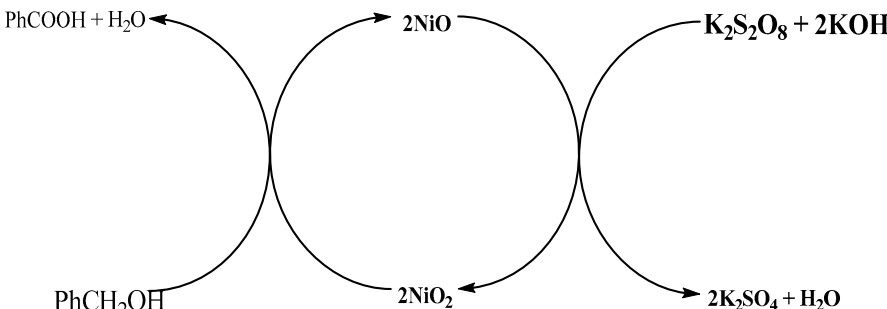

**Scheme 2.** Catalytic cycle of the oxidation of benzyl alcohol to benzoic acid.

## 3. Materials and Methods

All chemicals were purchased from Sigma-Aldrich (Berlin, Germany) ($NiSO_4 \cdot 6H_2O$, $NiCl_2 \cdot 2H_2O$, $Ni(CH_3COO)_2 \cdot 2H_2O$, $K_2S_2O_8$, KOH, benzyl alcohol (BzOH), and *p*-substituted benzyl alcohols (R-BzOH; R = $CH_3$, $OCH_3$, OH, $NH_2$, CHO, CN, $NO_2$, and $F_3C$)). The Fourier Transmission Infrared (FTIR) spectrum of $NiO_2$ was recorded through Alpha-Bruker FTIR spectrophotometer (model no. 200695, Berlin, Germany). The X-ray diffraction (XRD) pattern of $NiO_2$ with Cu K$\alpha$ radiation was obtained using PANalytical X'pert Pro X-ray diffractometer (DY 3190, Tokyo, Japan). Field Emission Scanning Electron Microscope (FESEM) Micrograph of $NiO_2$ was obtained by Carl Zeiss FESEM (SUPRA, 55VP, Berlin, Germany). Energy Dispersive X-ray (EDX) spectrum of $NiO_2$ was obtained by EDX detector of Oxford instruments (America) attached with FESEM. Transmission Electron Microscope (TEM) image of $NiO_2$ was taken by Jeol TEM (JEM-2100, Tokyo, Japan). $^1$H NMR spectra were recorded on a Bruker DPX (600 MHz, London, UK)

### 3.1. Preparation of Nickel Peroxide Nanoparticles (NPNPs)

The nickel peroxide (NPNPs) were prepared using the co-precipitation method [24,25]. In a typical experiment, a solution of $NiSO_4 \cdot 6H_2O$ (13.15 g, 0.05 mol) in 75 mL of $H_2O$ was added to the solution of $K_2S_2O_8$ (13.52 g, 0.05 mol) in 75 mL of $H_2O$ with stirring. To this mixture, 150 mL of 2.0 M KOH (16.8 g, in 150 mL $H_2O$) was added in portions within one hour with continual stirring for a further 2 h. A black fine precipitate of nickel peroxide (NPNPs) was formed and collected by filtration, then thoroughly washed with deionized water and dried at 60 °C (yield: 90%).

### 3.2. General Procedure for the Oxidation of Alcohol

Alcohol (10 mM) and NiSO$_4$·6H$_2$O (0.2 mM) were added to a 250 mL flat-bottomed flask containing 100 mL of 1.0 M KOH. The reaction mixture was stirred for the 10 min, K$_2$S$_2$O$_8$ (20 mM) was added in three portions, then the reaction mixture was stirred for three hours. The black color of the nickel peroxide appeared then gradually disappeared, then the reaction mixture was acidified with 10 mL of 5% HCl, extracted by diethyether (3 × 10 mL) to remove the unreacted alcohol, and filtered to collect the produced acid. The IR spectrum, melting point, and $^1$HNMR results were measured where appropriate and compared with authentic samples. The analysis for the produced acids is listed below.

1.  Benzoic acid

    White solid; mp 121–122 °C (122 °C) [20]. IR: 3405–2103, 1682, 1610, 1572, 1463, 1432, 1333, 1282 cm$^{-1}$. $^1$H NMR (400 MHz, CDCl$_3$): δ = 12.27 (br s, 1H, COOH), 8.43 (d, *J* = 7.1 Hz, 2H, Ar-H), 7.61 (t, *J* = 7.4 Hz, 1H, Ar-H), 7.47 (t, *J* = 7.8 Hz, 2H, Ar-H).

2.  4-Methylbenzoic acid

    Pale yellow solid; mp 180–181.2 °C (180–182 °C) [20]. IR: 3400–2295, 1666, 1621, 1579, 1526, 1427, 1330, 1286 cm$^{-1}$. $^1$H NMR (400 MHz, dmso-d$_6$): δ = 12.73 (br s, 1H, COOH), 7.67 (d, *J* = 8.0 Hz, 2H, Ar-H), 7.71 (d, *J* = 8.0 Hz, 2H, Ar-H), 2.58 (s, 3 H, CH$_3$).

3.  4-Methoxybenzoic acid

    Pale yellow solid; mp 181.6–184 °C (182–184 °C) [20]. IR (neat): 3310–2105, 1675, 1610, 1571, 1512, 1420, 1291, 1253,1172, 1160, 1123, 1100, 1020 cm$^{-1}$. $^1$H NMR (400 MHz, dmso-d$_6$): δ = 12.60 (br s, 1H, COOH), 7.93 (d, *J* = 8.8 Hz, 2H, Ar-H), 7.05 (d, *J* = 8.8 Hz, 2H, Ar-H), 3.82 (s, 3H, CH$_3$).

4.  4-Chlorobenzoic acid

    White solid; mp 238.0–239.0 °C (239–240 °C) [20]. IR: 3300–2200, 1678, 1591, 1574, 1492, 1423, 1400, 1320, 1305, 1176 cm$^{-1}$. $^1$H NMR (400 MHz, dmso-d$_6$): δ = 13.32 (br s, 1H, COOH), 7.78 (d, *J* = 8.8 Hz, 2H, Ar-H), 7.68 (d, *J* = 8.4 Hz, 2H, Ar-H).

5.  4-Nitrobenzoic acid

    Pale yellow solid; mp 138–139 °C (139 °C) [20]. IR: 3405–2210, 1709, 1611, 1571, 1517, 1475, 1418, 1343 cm$^{-1}$. $^1$H NMR (400 MHz, dmso-d$_6$): δ = 13.60 (br s, 1H, COOH), 8.43 (d, *J* = 8.0 Hz, 2H, Ar-H), 8.38 (d, *J* = 8.0 Hz, 2H, Ar-H).

6.  4-Cyanobenzoic acid

    Pale yellow solid; mp 221.0–222 °C (219–221 °C) [20]. IR (neat): 3410–2300, 2221, 1675, 1620, 1560, 1439, 1325, 1276, 1173, 1122, 1126, 1030 cm$^{-1}$. $^1$H NMR (400 MHz, dmso-d$_6$): δ = 13.60 (br s, 1H, COOH), 8.13 (d, *J* = 8.0 Hz, 2H, Ar-H), 8.03 (d, *J* = 8.4 Hz, 2H, Ar-H).

7.  4-Trifluoromethylbenzoic acid

    White solid; mp; 220.2–222.2 °C (220–222 °C) [20]. IR (neat): 3410–2105, 1684, 1583, 1515, 1420, 1310, 1285, 1130, 1125, 1121, 1055, 1028 cm$^{-1}$. $^1$H NMR (400 MHz, dmso-d$_6$): δ = 13.54 (br s, 1H, COOH), 8.16 (d, *J* = 8.0 Hz, 2H, Ar-H), 7.89 (d, *J* = 8.0 Hz, 2H, Ar-H).

8.  Piperonylic acid

    Colorless needles; mp 224–226 °C (228–232 °C) [20]. IR (neat): 3100, 2900, 2850, 1620, 1540, 1455, 1410, 1230, 1153, 1110, 1113, 1015 cm$^{-1}$. $^1$H NMR $^1$H NMR (250 MHz, CDCl3) 7.77 (s, 1H, COOH), 7.50 (dd, 1H, *J* = 8 and 2 Hz, Ar), 6.22 (d, 1H, *J* = 2 Hz, Ar).

## 4. Conclusions

We introduced a straightforward and efficient catalytic method for the conversion of benzyl alcohol and some para-substituted benzyl alcohols to their corresponding carboxylic acids using NiSO$_4$·6H$_2$O (0.2 mM)/K$_2$S$_2$O$_8$ (20 mM) in 100 mL of 1.0 M KOH and NiSO$_4$·6H$_2$O (0.2 mM)/KBrO$_3$ (20 mM) in 100 mL of 1.0 m K$_2$CO$_3$. The advantages of this

catalytic process are the use of non-toxic and inexpensive materials; the mild reaction conditions; the simple, safe procedure; and the short reaction times. The yields and turnover are good. This protocol can be extended for the catalytic oxidation of other organic substrates such as aromatic amines and secondary alcohols, which is currently under investigation in our laboratory.

**Author Contributions:** Conceptualization, A.G.F.S.; methodology, M.M.A.H.S.; software, N.A.E.-G.; validation, N.A.E.-G.; writing—original draft, A.G.F.S.; writing—review and editing, T.A.Y., M.M.A.-K. and S.H.K. All authors have read and agreed to the published version of the manuscript.

**Funding:** This research was supported by the Deanship of Scientific Research at Imam Mohammed Ibn Saudi Islamic University through research group no. RG-21-09-80.

**Data Availability Statement:** Not applicable.

**Conflicts of Interest:** The authors declare no conflict of interest.

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
