# Peer review of "Green Catalytic Conversion of Some Benzylic Alcohols to Acids by NiO2 Nanoparticles (NPNPs) in Water"

_catalysts, doi:10.3390/catal13040645_

Round 1

Reviewer 1 Report

Comments to the Author

The manuscript (catalysts-2276119) titled ‘Green catalytic conversion of some benzylic alcohols to acids by NiO2 nanoparticles (NPNPs) in water’ has been carefully reviewed. The manuscript is a good contribution to benzyl alcohol and para-substituted benzyl alcohols oxidation reaction under the relatively mild reaction conditions. The work elegantly applied NiO2 nanoparticles (NPNP), and investigated by various characterization techniques. This context of article is suitable to the scope of this journal. The work is comprehensive and the data is analyzed and presented. Overall, the study is deserved to be published in this journal, but minor revisions needed to be done as follows.

1. In this paper, K2S2O8 or KBrO3 and KOH are used to synthesize NiO2. What roles do they play in the synthesis process? And what is the difference between K2S2O8 and

KBrO3?

2. The XRD pattern of NiO2 powder is shown in Figure 3. However, some peaks do not explain what material they belong to. Please add the XRD standard card of NiO2 and other materials to explain the composition of catalyst more clearly and intuitively.

3. What is the formula for calculating the yield of reaction products? In addition, it was mentioned that when the reaction at 50 °C and 80 °C in the article, the benzoic acid yield decreased and unknown products appeared. As a result, whether there is excessive oxidation?

4. The authors noticed that the aromatic alcohols containing para substituted electron donation groups gave better yields than those having electron with drawing group. What are the possible reasons for this phenomenon? And please supply some convincing evidence to test.

5. The author tried to use different nickel salts (NiSO4, NiCl2 and Ni(CH3COO)2) to carry out the experiment, and the obtained results were also different. Please explain the reasons for the differences of the three anions in the reaction.

6. The languages should be carefully checked and polished. For example, Table 1 needs to adjust the table spacing as well as pay attention to the superscript and subscript of chemical formula in the abstract.

Author Response

Response to reviewer 1:

Thank you very much for these constructive comments; we highly appreciate your great effort and valuable time. Thanks very much for always giving me the way to the solutions and not just giving me comments. We accomplished all the suggested corrections, full of hope that these changes meet your expectations.

Comment 1: In this paper, K2S2O8 or KBrO3 and KOH are used to synthesize NiO2. What roles do they play in the synthesis process? And what is the difference between K2S2O8 and KBrO3?

Response: Explained in equation (1) and we added [ K2S2O8 or KBrO3 act as co-oxidants and react with NiSO4.6H2O in the presence of KOH as basic medium to generate NiO2 for example equation (1)]

NiSO4.6H2O + 4KOH + K2S2O8 → NiO2 + 3K2SO4 + 8H2O

Comment 2: The XRD pattern of NiO2 powder is shown in Figure 3. However, some peaks do not explain what material they belong to. Please add the XRD standard card of NiO2 and other materials to explain the composition of catalyst more clearly and intuitively.

Response: Two XRD standard cards were added in addition to indexing of the observed sharp peaks.

Comment 3: What is the formula for calculating the yield of reaction products? In addition, it was mentioned that when the reaction at 50 °C and 80 °C in the article, the benzoic acid yield decreased and unknown products appeared. As a result, whether there is excessive oxidation?

Response: The formula for calculating the yield of reaction products: Y (%) = number of moles of produced acid× 100/number of moles of alcohol. It was added as footnote of Table 1. Also, we addwd [The unknown products are mixture of unidentified organic product it comes from reaction of unreacted alcohol with the produced acid under these temperatures mentioned (50 °C and 80 °C) and actually it was difficult to isolate and identify]

Comment 4: The authors noticed that the aromatic alcohols containing para substituted electron donation groups gave better yields than those having electron with drawing group. What are the possible reasons for this phenomenon? And please supply some convincing evidence to test.

Response: Sorry for this mistake it was explained as [The possible reasons for this phenomenon; the electron-donating group activate the ring while thus it enhances oxidation of alcohol to corresponding acid while the electron-withdrawing deactive the phenyl ring so it retards the reaction and convincing evidence is the yield obtained with the first is than that obtained with the later ]

Comment 5: The author tried to use different nickel salts (NiSO4, NiCl2 and Ni(CH3COO)2) to carry out the experiment, and the obtained results were also different. Please explain the reasons for the differences of the three anions in the reaction.

Response: Sorry for this mistake it was illustrated as [The differences of the three anions in the reaction is the solubility of NiSO4.6H2O is better than the solubility of other salts and this comes from the higher yields of NiSO4.6H2O than with NiCl2.2H2O and Ni(CH3COO)2)2. 2H2O]

Comment 6: The languages should be carefully checked and polished. For example, Table 1 needs to adjust the table spacing as well as pay attention to the superscript and subscript of chemical formula in the abstract.

Response 6:  The spacing of Table1 is adjusted and the subscript of chemical formula are corrected.

Reviewer 2 Report

The manuscript described the catalytic activity of NiO2 nanoparticles for converting benzylic alcohols into the corresponding acids in aqueous media. Various characterization techniques support the study, which can be an excellent addition to developing a green methodology. Therefore, I recommend accepting it after revision.

General comments:

  1. Figures 1-3: label the peak values.
  2. Table 1: Provide appropriate footnotes with reaction conditions.
  3. All reported final compounds (acids) are known in the literature. Therefore, it would be nice if the author could perform the reaction which produced an unknown acid, and the results should be supported by at least FTIR, Mass, 1H, and 13C NMR data.

Author Response

Response to reviewer 2:

Thank you very much for your observations and opinions, which helped us greatly in improving our manuscript. We accomplished most of the suggested requests, full of hope that these changes meet your expectations.

Comment 1: Figures 1-3: label the peak values.

Response: done

Comment 2: Table 1. Provide appropriate footnotes with reaction conditions.

Response: The footnotes were added with reaction conditions.

Comment 3: All reported final compounds (acids) are known in the literature. Therefore, it would be nice if the author could perform the reaction which produced an unknown acid, and the results should be supported by at least FTIR, Mass, 1H, and 13C NMR data.

Response: Piperonyl alcohol was oxidized piperonylic acid by this catalytic system and the results were added in Table1 (contd).